# An investigation into waste handler's knowledge of management of isolation waste: A case study of Dr George Mukhari Academic Hospital, Gauteng, South Africa

**Mmatlou Ouma Moloisi**[1]**, Stanley Chibuzor Onwubu**[2]*

**1** Department of Community Health, Durban University of Technology, Durban, South Africa, **2** Department of Chemistry, Durban University of Technology, Durban, South Africa

* stanleyo@dut.ac.za

## Abstract

### Background

The proper management of isolation waste is of utmost importance in healthcare facilities to prevent the spread of infections and protect both healthcare workers and the general public. This study investigated waste handlers' knowledge of the management of isolation waste at Dr. George Mukhari Academic Hospital in Gauteng, South Africa.

### Methods

A survey was conducted to assess waste handlers' understanding of waste types, colour codes, safety precautions, and awareness of internal policies related to isolation waste management.

### Results

The study found that the majority of waste handlers demonstrated a good understanding of waste types generated in the isolation unit, including sharps waste, human tissue waste, infectious waste, and general waste. They also correctly identified examples of sharp waste, such as injections, blades, glass slides, and needles. Additionally, most respondents were aware of the colour code used for representing infectious waste as "yellow" and "red." The study revealed a statistically significant association between waste handlers' age and their knowledge of isolation waste, suggesting that age may influence their understanding of waste management practices. Furthermore, experience was found to be significantly associated with waste handlers' knowledge of the health-hazardous nature of isolation waste. While the majority of waste handlers recognized the importance of wearing protective clothing and correctly marking isolation waste, some respondents were not aware of the internal policy for waste handling such as guidelines and protocols specific to the segregation, packaging, labeling, and disposal of waste generated within the isolation units.

**Data Availability Statement:** Data are available within the manuscript.

**Funding:** The author(s) received no specific funding for this work.

**Competing interests:** The authors have declared that no competing interests exist.

## Conclusion and contribution

These findings highlight the importance of continuous training, targeted education, and policy dissemination to ensure effective waste management and adherence to safety protocols among waste handlers.

## 1. Introduction

Healthcare facilities, including hospitals, generate various types of medical waste, some of which can pose significant health and environmental risks if not managed properly [1]. Isolation waste, in particular, is a category of medical waste that requires special attention due to its potential to carry infectious agents [2]. These waste materials are generated from patients with contagious diseases or conditions, such as COVID-19, tuberculosis, or other communicable illnesses [3]. Proper management of isolation waste is crucial to prevent the transmission of infections among healthcare workers, patients, visitors, and waste handlers themselves [4]. Inadequate waste management practices can lead to the spread of infections both within the healthcare facility and in the broader community [2]. Particularly, improper disposal of isolation waste presents significant risks to both healthcare workers and the environment. For instance, inadequate disposal of sharps waste, including needles and syringes, can lead to needlestick injuries among healthcare workers. Such injuries pose a serious risk of exposure to bloodborne pathogens such as HIV, hepatitis B, and hepatitis C [5]. Similarly, improper disposal of isolation waste, such as dumping or inadequate landfilling, can have detrimental effects on the environment. Contaminating soil and groundwater with hazardous chemicals and pathogens from isolation waste poses risks to ecosystems and human health [6].

Dr George Mukhari Academic Hospital, situated in Gauteng, South Africa, serves as a significant tertiary medical institution in the region. As a facility providing healthcare services to a diverse patient population, including those with infectious diseases, the proper handling and disposal of isolation waste are paramount to ensure the safety and well-being of all individuals involved. The management of isolation waste involves several critical steps, including waste segregation, proper containment, safe transportation, and appropriate disposal [7].

Healthcare waste handlers play a pivotal role in executing these tasks [8]. They include waste management staff, nurses, and other healthcare professionals directly involved in the collection, handling, and disposal of isolation waste. These individuals must possess adequate knowledge and understanding of isolation waste management protocols and guidelines to minimize the risk of infection transmission [9].

In South Africa, the management of Healthcare Risk Waste (HCRW) is subject to regulation by various legislation. Key among these are the National Health Act No. 61 of 2003, the National Environmental Management Act No. 107 of 1998, the National Environmental Management Waste Act No. 59 of 2008, and National Core Standards [10]. These regulations, based on the Cradle-to-Grave principle, establish comprehensive protocols governing the handling, storage, collection, transportation, treatment, and disposal of medical waste (Fig 1). The primary objective of these regulations is to minimize the risks associated with Healthcare Risk Waste, thereby reducing the likelihood of disease outbreaks and the transmission of infections within healthcare settings. In principle, HCRW has to be transferred from point of generation to point of treatment before disposal to the landfill site [11]. Despite these regulations, compliance remains a challenge in many hospitals across South Africa. While specific national and

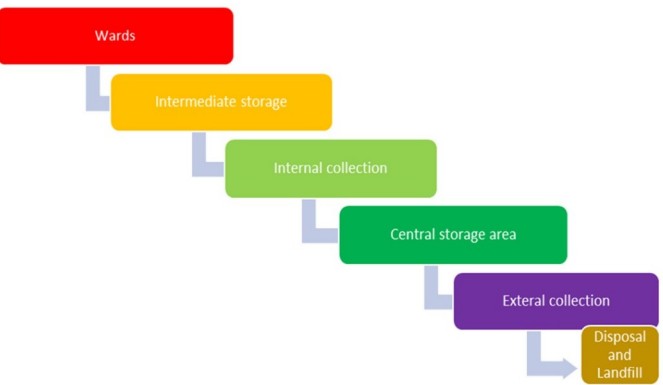

**Fig 1. Cradle to grave approach for HCRW management in SA (authors own creation).**

regional guidelines exist for healthcare waste management, there may be variations in knowledge, awareness, and adherence to these guidelines among waste handlers in different healthcare settings [10]. Investigating the adherence to local waste management regulations and guidelines, particularly in the context of isolation waste, can shed light on the current state of waste handlers' knowledge and understanding regarding isolation waste management. While healthcare waste management practices in healthcare settings have been extensively studied [12–14], there is a dearth of research specifically focusing on waste handlers' knowledge regarding the management of isolation waste in South Africa. Understanding waste handlers' level of awareness, adherence to guidelines, and comprehension of waste segregation and disposal procedures is crucial to improving overall waste management practices and minimizing infection transmission risks.

Furthermore, it is critical to note that each healthcare facility may encounter unique challenges related to waste management [15]. Dr George Mukhari Academic Hospital, being one of the prominent medical institutions in Gauteng, may face context-specific issues that impact waste handlers' practices and compliance with isolation waste management protocols. Identifying these challenges and understanding their impact on waste management at this hospital can provide valuable insights for other similar healthcare facilities facing comparable issues. This study aimed to investigate the knowledge of waste handlers regarding the management of isolation waste at Dr George Mukhari Academic Hospital (DGMAH). This study through a comprehensive investigation at Dr George Mukhari Academic Hospital will contribute to the body of knowledge related to healthcare waste management and provide actionable insights for enhancing isolation waste management practices.

## 2. Research methodology

### 2.1 Study site

This study was conducted at DGMAH which is a tertiary hospital providing national referral services, training of healthcare providers, conduct research, and care for highly infectious diseases or conditions that require isolation. The hospital is situated in the city of Tshwane, Gauteng province and has a bed capacity of 1652. Dr George Mukhari Academic Hospital also provides specialised healthcare services to patients who are referred by their local clinics and district and regional hospitals.

## 2.2 Study design

This was a descriptive study that followed a quantitative cross-sectional study design. This design was chosen, as it would provide information on the knowledge of waste handlers for the management of waste from isolation units at DGMAH.

## 2.3 Study procedure

Permission to conduct the study at Dr George Mukhari Academic Hospital was granted by the Department of Health through the Health Research Unit, the body responsible for approving all research studies conducted within hospital settings. Subsequently, permission to contact participants was obtained from the hospital management, following the presentation of approval from the NHRD. The hospital management played a crucial role in participant recruitment by disseminating information about the study to the heads of departments and waste handlers in their respective departments. Participants were purposively selected from waste handlers stationed in hospital wards, specifically those responsible for handling isolation waste, including its removal, packaging, and sealing. Unlike regular medical waste, isolation waste remains within the room until removed by an external service provider, reducing the likelihood of other waste handlers, such as those responsible for internal collection, coming into contact with such waste.

A total of 34 waste handlers participated in the study, all of whom are involved in waste collection at DGMAH. Recruitment took place between November 11, 2019, and December 4, 2019. Before participating in the study, participants were asked to explain their daily duties to the researcher. This process was crucial for identifying individuals who met the study criteria.

Data collection was conducted using a structured questionnaire comprising close-ended questions. The use of closed questions facilitated easy processing of answers, enhanced comparability, and facilitated the presentation of relationships between variables, allowing for comparisons between respondents. The questionnaire consisted of four parts, and data variables were categorized accordingly as shown in Fig 2.

To assess the knowledge of waste handlers on handling isolation waste, a rating scale was used whereby responses were rated from 0–2, with 0 indicating very poor, 1 being good and 2 being very good. To respond to questions on the usage of hygiene, PPE, safety precautions and training, respondents had to select the correct answer and three or five options were listed, some questions required respondents to give a yes/no/ do not know response. The questionnaire was both in Sepedi and English to ensure that the participants understood the questions clearly.

## 2.4 Data analysis

The raw data gathered was properly organised and carefully checked for any errors and inconsistencies. Furthermore, a unique code was used to identify each questionnaire. The data were summarised on the spreadsheet, entered into SPSS version 27 and analysed with the assistance of a statistician. Results were presented as descriptive statistics in the form of tables, graphs and cross-tabulations. Inferential statistics such as Chi-square were used, which were



**Fig 2. Flowchart of data questionnaire categoriztion (authors own creation).**

interpreted using p-values. A p-value of 0.05 was considered significant. The variables of interest included the level of knowledge of different types of waste from the isolation unit, usage of PPE, and risks of handling isolation waste. Frequency distribution was calculated for knowledge on usage of PPE, staff training, handling spillages and knowledge on hand hygiene. All questions whereby a respondent had the option of scoring more than one correct answer were scored as 2 if the respondent managed to get more than 1 answer correct. For all other questions that were responded to correctly and positively, a score of 1 was given. All negative and incorrect answers were scored as 0. All the 2s and 1s were summed up and the key findings were expressed utilizing graphs and tabulation. The chi-square test of independence was used to determine any relationship between the level of knowledge of a waste handler and the profile of the waste handlers.

## 2.5 Ethical consideration

Ethical clearance for this study was obtained from the Institutional Research Ethics Committee at Durban University of Technology, under the assigned IREC number 139/19. Subsequently, permission to conduct the study and contact participants was sought and granted from the Gauteng Department of Health. Participation in the study was entirely voluntary, and participants had the right to withdraw at any point without facing any consequences. Those who agreed to take part were provided with consent forms and letters of information detailing the study's purpose, procedures, and their rights as participants. To ensure confidentiality, the researcher refrained from using participants' names in the report, and no information or data that could potentially identify respondents was included. The collected information was solely used for the research report and was not shared for any other purpose. Participants were also assured that their involvement in the study would not result in harm or injury. Moreover, the potential benefits of the study were explained, emphasizing its contribution to improving isolation waste management practices in healthcare facilities.

## 3. Results

### 3.1 Socio-demographic characteristics

The socio-demographic characteristic of the waste handlers is given in Table 1. The gender distribution showed 85.3% females and 14.7% males. In terms of age groups, half of the respondents (50%) were between 40–49. Education-wise, most of the respondents (47.1%) completed high school while those who had some high schooling makes up 38.2% of sample. In terms of their experience as a waste handler, most of the respondents (47.1%) had 6–11 years.

### 3.2 Waste handlers' knowledge of isolation waste

Table 2 focuses on the knowledge of waste handlers with the types of waste generated, labelling and health hazards associated with isolation waste. When asked to indicate the type of waste generated in an isolation unit, the majority (n = 25; 73.5%) of the respondents correctly indicated the answers as sharps waste, human tissue waste, infectious waste, and general waste. Similarly, the majority (n = 26; 76.5%) correctly identified injections, blades, glass slides, and needles as an example of sharp waste.

In terms of the knowledge of the colour code used to represent infectious waste, the majority of respondents (n = 30; 88.2%) correctly indicated "yellow" and "red". Regarding the waste handlers' knowledge of different types of infectious waste in the isolation ward, the majority of the respondents (n = 25; 73.5%) correctly indicated bandages, used cotton wools, dirty nappies, and used gloves.

**Table 1. Respondents' demographic variables.**

| Variable | Frequency (n = 34) | Percentage (%) |
|---|---|---|
| Gender | | |
| Male | 5 | 14.7 |
| Female | 29 | 85.3 |
| Age group | | |
| Below 20 | 1 | 2.9 |
| 21–39 | 8 | 23.5 |
| 40–49 | 17 | 50 |
| 50 and below | 8 | 23.5 |
| Education | | |
| Some primary schooling | 3 | 8.8 |
| Completed primary school | 2 | 5.9 |
| Some high schooling | 13 | 38.2 |
| Completed high school | 16 | 47.1 |
| Years of experience | | |
| Below 5 | 12 | 35.3 |
| 6–11 | 16 | 47.1 |
| 12–23 | 5 | 14.7 |
| 24 and above | 1 | 2.9 |

In terms of the colour used to represent the container where the infectious waste was disposed, the majority (n = 27; 79.4%) got only one correct answer whilst 7 (20.6%) correctly indicated "yellow" and "red". When asked whether isolation waste must be marked or labelled as isolation waste, an overwhelming majority (n = 32; 94.1%) correctly indicated "yes" whilst only 2 (5.9%) respondents incorrectly answered the question. In terms of the respondent's knowledge of the hazardous nature of isolation waste, the majority of respondents (n = 32; 97%) correctly indicated that waste in the isolation unit was very dangerous and could spread very dangerous diseases while only 1 (3.0%) incorrectly thought otherwise.

Furthermore, and with respect to the knowledge of waste handlers on safety precautions to follow when handling isolation waste, the majority of the respondents (n = 28; 82,4%) correctly indicated that the waste must be correctly sealed, labelled, and packaged.

**Table 2. Knowledge of waste handlers on different types of waste generated in isolation unit.**

| Statements | Knowledge of waste handlers on different types of waste generated in the isolation unit (n = 34) | | | | | |
|---|---|---|---|---|---|---|
| | All correct response | | One correct response | | Incorrect response | |
| | Frequency | % | Frequency | % | Frequency | % |
| Knowledge of waste handlers with the types of waste generated, labelling and health hazards associated with isolation waste | 25 | 73.5 | 8 | 23.5 | 1 | 2.9 |
| Knowledge of sharps waste | 26 | 76.5 | 8 | 23.5 | 0 | 0 |
| Knowledge on colour code used for sharps waste. | 30 | 88.2 | 0 | 0 | 4 | 11.8 |
| knowledge of different types of infectious waste in the isolation ward | 25 | 73.5 | 1 | 2.9 | 8 | 23.5 |
| Knowledge on colour code for isolation waste | 7 | 20.6 | 27 | 73.5 | 0 | 0 |
| Isolation waste must be labelled or marked with the term isolation waste | 32 | 94.1 | 0 | 0 | 2 | 5.9 |
| Dangerous nature of waste in the isolation unit | 33 | 97.1 | 0 | 0 | 1 | 2.9 |
| Safety precaution when removing waste from the isolation unit | 28 | 82.4 | 6 | 17.6 | 0 | 0 |

**Table 3. Association between demographic data and knowledge of isolation waste.**

| Knowledge on isolation unit waste | | Age | Gender | Education | Experience |
|---|---|---|---|---|---|
| Type of waste is generated in the isolation unit | Chi-square | 10.666 | 0.237 | 8.857 | 6.282 |
| | Df | 6 | 2 | 6 | 8 |
| | Sig. | 0.099 | 0.888 | 0.182 | 0.616 |
| Example of sharps waste | Chi-square | 4.986 | 0.884 | 4.965 | 2.877 |
| | Df | 3 | 1 | 3 | 4 |
| | Sig. | 0.173 | 0.347 | 0.174 | 0.579 |
| Colour used for sharps waste | Chi-square | 2.054 | 0.782 | 0.839 | 6.384 |
| | Df | 3 | 1 | 3 | 4 |
| | Sig. | 0.561 | 0.377 | 0.840 | 0.172 |
| Isolation waste | Chi-square | 18.053 | 0.994 | 12.727 | 6.494 |
| | Df | 6 | 2 | 6 | 8 |
| | Sig. | 0.006 | 0.608 | 0.048 | 0.592 |
| Colour used to represent isolation waste | Chi-square | 2.069 | 1.351 | 1.536 | 7.088 |
| | Df | 3 | 1 | 3 | 4 |
| | Sig. | 0.558 | 0.245 | 0.674 | 0.131 |
| Marking or labelling as isolation waste | Chi-square | 1.195 | 0.366 | 3.433 | 2.692 |
| | Df | 3 | 1 | 3 | 4 |
| | Sig. | 0.754 | 0.545 | 0.330 | 0.611 |
| Health hazardous nature of isolation waste | Chi-square | 3.830 | 5.775 | 1.587 | 1.805 |
| | Df | 3 | 1 | 3 | 3 |
| | Sig. | 0.280 | 0.016 | 0.662 | 0.614 |
| Safety precaution on removing waste from isolation unit | Chi-square | 7.693 | 0.013 | 6.732 | 5.215 |
| | Df | 3 | 1 | 3 | 3 |
| | Sig. | 0.053 | 0.909 | 0.081 | 0.157 |
| Correct bagged from prevention of disease transmission | Chi-square | 2.576 | 2.011 | 5.077 | 2.848 |
| | Df | 3 | 1 | 3 | 3 |
| | Sig. | 0.462 | 0.156 | 0.166 | 0.416 |

Table 3 shows the Chi-square tests between the knowledge levels and various demographic and professional factors. The data analysis showed no statistically significant association between the type of waste generated in the isolation unit and the waste handlers' age, gender, education, or experience ($p > 0.05$). This suggests that the knowledge of waste handlers regarding the types of waste generated in the isolation unit was not significantly influenced by these demographic and professional factors.

Similarly, there was no significant association between the waste handlers' ability to provide an example of sharps waste and their age, gender, education, or experience ($p > 0.05$). The choice of colour used for sharps waste disposal did not show any significant association with waste handlers' age, gender, education, or experience ($p > 0.05$). However, concerning knowledge of isolation waste in general, the data analysis revealed a statistically significant association with waste handlers' age ($p = 0.006$). This suggests that waste handlers' understanding of isolation waste might vary depending on their age.

There was no significant association between the colour used to represent isolation waste and the waste handlers' age, gender, education, or experience ($p > 0.05$). Similarly, the marking or labelling of isolation waste did not show any significant association with waste handlers' age, gender, education, or experience ($p > 0.05$). The data analysis indicated a significant association between waste handlers' knowledge of the health-hazardous nature of isolation waste

Table 4. Waste handlers understanding of isolation waste management.

| Statements | Knowledge of waste handlers on different types of waste generated in the isolation unit (n = 34) | | | | | |
|---|---|---|---|---|---|---|
| | All correct response | | One correct response | | Incorrect response | |
| | Frequency | % | Frequency | % | Frequency | % |
| Understanding the means of spreading disease in the isolation unit | 25 | 73.5 | 7 | 20.6 | 2 | 5.9 |
| Protective clothing is important for my protection against diseases as a waste handler. | 34 | 100 | 0 | 0 | 0 | 0 |
| Protective clothing should always be worn when handling isolation waste. | 34 | 100 | 0 | 0 | 0 | 11.8 |
| Protective clothing should always be worn when handling isolation waste. | 34 | 100 | 0 | 0 | 0 | 0 |
| Correct step to take when stick with needle injury | 15 | 44.1 | 18 | 52.9 | 1 | 2.9 |
| Number of times waste handlers are trained in a year | 29 | 85.3 | 0 | 0 | 5 | 14.7 |

and their experience (p = 0.016). This suggests that waste handlers with more experience may have a better understanding of the potential health risks associated with isolation waste. There was no significant association between the knowledge of safety precautions on removing waste from the isolation unit and waste handlers' age, gender, education, or experience (p > 0.05). The correct bagging for the prevention of disease transmission did not show any significant association with waste handlers' age, gender, education, or experience (p > 0.05).

### 3.3 Understanding the means of spreading disease through isolation waste

Table 4 presents the knowledge of waste handlers at Dr George Mukhari Academic Hospital regarding different aspects of isolation waste management. Out of the 34 waste handlers surveyed, 25 of them (73.5%) demonstrated a correct understanding of the means of spreading disease in the isolation unit. They were able to identify the possible routes of disease transmission within the isolation unit. However, there were 7 respondents (20.6%) who provided only one correct response despite being allowed to select multiple answers. This tendency may indicate a potential limitation in the depth of their knowledge. Furthermore, 2 participants (5.9%) provided incorrect responses to the question.

All 34 waste handlers (100%) correctly recognized the importance of wearing protective clothing for their protection against diseases when handling isolation waste. This indicates a high level of awareness among the waste handlers regarding personal protective equipment (PPE) usage. All 34 waste handlers (100%) provided the correct response, stating that protective clothing should always be worn when handling isolated waste. This indicates a strong understanding of the importance of using protective clothing consistently during waste handling activities. Fifteen waste handlers (44.1%) responded correctly regarding the correct steps to take when faced with a needlestick injury. However, 18 respondents (52.9%) provided one correct response, indicating that some waste handlers might be unsure or have limited knowledge about the appropriate actions to take in such situations. Additionally, one respondent (2.9%) provided an incorrect response.

### 3.4 Training and knowledge about policies on isolation waste

Table 5 presents the training and knowledge about policies regarding isolation waste. In terms of the statement "I am aware of internal policy for handling waste from the isolation unit", out of the 32 that responded, 23 (67.6%) respondents indicated they were aware of the internal policy for handling waste in the isolation units whilst 11 (32.4%) respondents were not aware of such a policy. Regarding whether the respondents (n = 34) have been trained on healthcare waste isolation waste management practices, the majority (n = 29; 85.3%) of respondents

**Table 5. Waste handlers' knowledge about policies on isolation waste.**

| Statements | Training and knowledge about policies on isolation waste (n = 34) | | | |
|---|---|---|---|---|
| | Aware | | Not Aware | |
| | Frequency | % | Frequency | % |
| I am aware of internal policy for handling waste from the isolation unit. | 23 | 67.6% | 11 | 32.4% |
| I have been trained on healthcare waste isolation waste management practices | 29 | 85.3 | 5 | 14.7 |
| Number of times waste handlers are trained in a year | 29 | 85.3 | 5 | 14.7 |
| Training covered the duties done by the waste handlers | 29 | 85.3 | 5 | 14.7 |
| Training helped to change the understanding of isolation waste practices | 29 | 85.3% | 5 | 14.7 |

agreed they had been trained while only 5 (14.7%) respondents indicated they had not undergone any training.

Out of the 34 waste handlers, 29 of them (85.3%) were aware of the number of times they were trained in a year regarding waste management and safety protocols. On the other hand, 5 respondents (14.7%) were uncertain or did not have knowledge of the frequency of their training sessions. In terms of whether the training covered the duties done by the respondents, the majority (n = 29; 85.3%) indicated that the training covered the duties done by them while 5 (14.7%) did not think so. When asked if the training the waste handlers underwent did help change their understanding of isolation waste practices. Among those who responded (n = 34), the majority (n = 29; 85.3%) indicated the training helped change their understanding of isolation waste while a few (n = 5; 14.7%) disagreed.

Table 6 presents the Chi-square tests waste handlers' knowledge of isolation waste management and various demographic and professional factors, including age, gender, education, and experience. The Chi-square test indicated that there was no statistically significant association between waste handlers' awareness of the internal policy for handling waste in the isolation unit and their age, gender, education, or experience (p > 0.05). This suggests that knowledge of the internal policy is not significantly influenced by these demographic and professional factors. The results also showed no significant association between waste handlers' knowledge of training on healthcare waste isolation waste management practices and their age, gender,

**Table 6. Association between demographic data and knowledge on isolation unit waste polices.**

| Knowledge of isolation waste | | Age | Gender | Education | Experience |
|---|---|---|---|---|---|
| Awareness of internal policy for handling waste in the isolation unit | Chi-square | 6.445 | 3.344 | 6.585 | 5.779 |
| | Df | 3 | 1 | 3 | 3 |
| | Sig. | 0.092 | 0.067 | 0.086 | 0.123 |
| Training on healthcare waste isolation waste management practices | Chi-square | 4.392 | 0.305 | 2.797 | 8.923 |
| | Df | 3 | 1 | 3 | 3 |
| | Sig. | 0.222 | 0.581 | 0.424 | 0.030 |
| Knowledge of the number of times waste handlers are trained in a year. | Chi-square | 4.688 | 0.108 | 2.604 | 9.275 |
| | Df | 3 | 1 | 3 | 3 |
| | Sig. | 0.196 | 0.743 | 0.457 | 0.026 |
| Training cover the duties | Chi-square | 4.688 | 0.108 | 2.604 | 9.275 |
| | Df | 3 | 1 | 3 | 3 |
| | Sig. | 0.196 | 0.743 | 0.457 | 0.026 |
| Training helping to change the understanding of isolation waste practices | Chi-square | 5.282 | 0.066 | 3.380 | 11.836 |
| | Df | 3 | 1 | 3 | 3 |
| | Sig. | 0.152 | 0.797 | 0.337 | 0.008 |

education, or experience (p > 0.05). This implies that the level of knowledge regarding training received does not vary significantly based on these factors.

Similarly, the Chi-square test did not reveal any significant association between waste handlers' knowledge of the number of times they receive training in a year and their age, gender, education, or experience (p > 0.05). The analysis indicated no statistically significant association between waste handlers' knowledge of whether their training covers their duties and their age, gender, education, or experience (p > 0.05). The Chi-square test showed that there was a statistically significant association between waste handlers' knowledge of whether their training helped change their understanding of isolation waste practices and their experience (p = 0.008). This suggests that more experienced waste handlers may have found the training to be more effective in improving their understanding of isolation waste management practices compared to less experienced handlers.

## 4. Discussion

Proper management and disposal of sharp waste are crucial in minimising the negative effects on the health of the community and the environment [16]. The study aimed to assess waste handlers' knowledge and understanding of isolation waste management at Dr. George Mukhari Academic Hospital in Gauteng, South Africa. Waste handlers knowledge and understanding of solation waste were assessed using a structured questionnaire. All questions that allowed respondents to select more than one correct answer were assigned a score of 2 if the respondent provided multiple correct answers. For questions with single correct responses answered correctly, a score of 1 was awarded. Incorrect or negative responses were scored as 0. The findings revealed that a majority of waste handlers demonstrated a good understanding of the types of waste generated in the isolation unit, as well as the specific examples of sharp waste, such as injections, blades, glass slides, and needles. This is practically important as the understanding of waste handlers in the handling of isolation waste could contribute to a safer and healthier environment within the healthcare facility. Besides, the understanding the types of waste, especially sharp waste, ensures that waste handlers can take appropriate safety precautions to prevent injuries and potential exposure to harmful pathogens. Proper handling and disposal of sharp waste reduce the risk of needlestick injuries and transmission of bloodborne infections such as HIV, hepatitis B, and hepatitis C. This aligns with Sutrisno and Meilasari [17] that effective management of the waste generated is of vital importance to curb the spread of disease.

Moreover, the majority of respondents correctly identified the colour code used to represent infectious waste as "yellow" and "red," indicating their awareness of the importance of proper waste segregation. The findings is consistent with Senekane and Masimula [11] that 90% medical practitioners generated waste and disposed of into bins lined with red plastics, 2.5% disposed of into yellow plastic bags. According to Andeobu, Wibowo [18], adequate waste categorization and disposal practices have implications for environmental protection. It is reasonable to agree that by correctly segregating and disposing of waste, the facility minimizes the potential negative impacts on the environment, such as soil and water contamination [19]. This aligns with broader sustainability goals and waste reduction initiatives of South Africa [20]. The proper categorization and segregation of different types of waste, such as hazardous and non-hazardous waste, and the implementation of appropriate disposal methods, are highly significant for South Africa. These practices can minimize the release of harmful pollutants into the environment, which is crucial for protecting ecosystems and safeguarding human health [18]. In addition, waste categorization and disposal practices are essential

components of South Africa's efforts to achieve environmental protection and sustainability goals, as outlined in the legislative framework [20].

The study further highlighted the importance of proper labelling and marking of isolation waste, as an overwhelming majority of respondents correctly indicated that isolation waste must be labelled as such. Proper labelling ensures clear communication and identification of hazardous waste, promoting safe handling and disposal practices [21]. Furthermore, the respondents' awareness of the hazardous nature of isolation waste was notable, with the vast majority correctly indicating that waste in the isolation unit is very dangerous and can spread dangerous diseases. This knowledge is crucial in instilling a sense of caution and responsibility among waste handlers, leading to better compliance with safety protocols [22].

Despite the positive findings regarding waste handlers' knowledge, continuous training programs should be implemented to reinforce proper waste management practices. Regular training sessions can help keep waste handlers updated on best practices and emerging guidelines, ensuring a consistently high level of knowledge [1]. The statistically significant association between age and knowledge of isolation waste suggests that age-specific training might be beneficial. Tailoring training programs to cater to the specific needs and learning styles of different age groups could improve overall understanding and adherence to waste management protocols.

Furthermore, waste handlers with more experience demonstrated a better understanding of the potential health risks associated with isolation waste. Given this finding, it is agreeable to say that utilising experienced waste handlers as mentors or educators could enhance knowledge dissemination among newer staff members. According to Mazorodze and Buckley [23], mentoring is effective disseminating knowledge as skills can be transferred from more experience workers to less experienced workers through mentoring. This is practically relevant as a substantial portion of waste handlers indicated that they were not aware of the internal policy for handling waste in the isolation unit. Efforts should, therefore, be made to improve the dissemination of policies and guidelines within the hospital to ensure consistent waste management practices. This is can be supported by Ezirim, Agbo [24] that health facilities should put standard operating procedures in place to guide day to day healthcare waste management operations.

While most waste handlers demonstrated knowledge of the appropriate steps to take in the event of a needlestick injury, there were some respondents who provided incorrect or partial responses. Specific training on responding to needlestick injuries can enhance preparedness and minimize potential risks [25]. This is vital as the finding showed that the majority of waste handlers found the training to be effective in changing their understanding of isolation waste practices. However, continuous evaluation and feedback mechanisms are essential to identify areas for improvement and to ensure training remains impactful. Although all waste handlers recognised the importance of wearing protective clothing, continuous reinforcement of the proper use of personal protective equipment is essential to maintain a safe working environment and prevent infection transmission.

## 5. Conclusion

The study focused on waste handlers' knowledge of isolation waste management at Dr. George Mukhari Academic Hospital in Gauteng, South Africa. Waste handlers demonstrated a commendable level of understanding, correctly identifying various waste types generated in the isolation unit and examples of sharp waste. Most waste handlers were aware of the color code used for infectious waste and recognized the importance of wearing protective clothing and correctly marking isolation waste.

However, the study revealed areas for improvement, including limited awareness of the internal policy for waste handling. The statistically significant association between age and knowledge of isolation waste suggested the potential benefits of targeted training for different age groups. Training initiatives could include on-the-job training sessions designed for healthcare workers and waste handlers to familiarize them with isolation waste management protocols, safety procedures, and regulatory requirements. Additionally, the provision of continuing education programs or professional development courses focusing on isolation waste management topics, such as infection control, hazardous waste handling, and personal protective equipment (PPE) usage, is essential. These initiatives aim to reinforce waste management practices and promote adherence to safety protocols.

Furthermore, utilizing experienced waste handlers as mentors can enhance knowledge dissemination among newer staff members. To achieve this, formal mentorship programs can be established, pairing experienced waste handlers with newer staff members based on job roles, areas of expertise, or specific needs identified during training.

## 6. Recommendations

Overall, the study recommends that Dr. George Mukhari Academic Hospital conduct regular training sessions and workshops for healthcare workers and waste handlers on proper waste segregation, handling, and disposal procedures. This includes providing education on infection control measures, instructing on the proper use of personal protective equipment (PPE), and ensuring compliance with regulatory requirements. Additionally, hospital management should regularly monitor and supervise waste management practices to ensure compliance with protocols and regulations. By following these recommendations, Dr. George Mukhari Academic Hospital can further improve waste management practices, ensure a safe working environment, and contribute to infection control and environmental protection.

## Acknowledgments

The author acknowledges the support provided by the management of George Mukhari Academic Hospital.

**Disclaimer:** The findings and content of this article are the authors views and not an official position of the institution or funder.

## Author Contributions

**Conceptualization:** Mmatlou Ouma Moloisi.

**Formal analysis:** Mmatlou Ouma Moloisi, Stanley Chibuzor Onwubu.

**Investigation:** Mmatlou Ouma Moloisi.

**Methodology:** Mmatlou Ouma Moloisi.

**Writing – original draft:** Mmatlou Ouma Moloisi, Stanley Chibuzor Onwubu.

**Writing – review & editing:** Mmatlou Ouma Moloisi, Stanley Chibuzor Onwubu.

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
