## [Decision Letter · Decision Letter 0]

8 Dec 2023

PONE-D-23-28839An investigation into waste handler’s knowledge of management of isolation waste: A case study of Dr George Mukhari Academic Hospital, Gauteng, South AfricaPLOS ONE

Dear Dr. Onwubu,

Thank you for submitting your manuscript to PLOS ONE. After careful consideration, we feel that it has merit but does not fully meet PLOS ONE’s publication criteria as it currently stands. Therefore, we invite you to submit a revised version of the manuscript that addresses the points raised during the review process.

**ACADEMIC EDITOR: Please, kindly see below.**

We look forward to receiving your revised manuscript.

Kind regards,

Charles Odilichukwu R. Okpala

Academic Editor

PLOS ONE

Journal Requirements:

Additional Editor Comments :

Please, kindly address the comments raised. Thanks

Reviewers' comments:

Reviewer's Responses to Questions

**Comments to the Author**

1. Is the manuscript technically sound, and do the data support the conclusions?

Reviewer #1: Partly

2. Has the statistical analysis been performed appropriately and rigorously? 

Reviewer #1: No

3. Have the authors made all data underlying the findings in their manuscript fully available?

Reviewer #1: No

4. Is the manuscript presented in an intelligible fashion and written in standard English?

Reviewer #1: Yes

5. Review Comments to the Author

Reviewer #1: General Comments

The paper presented information regarding the respondents' (represented by 34 respondents) awareness of waste handling and waste identification at Dr. George Mukhari Academic Hospital (DGMAH), Gauteng, South Africa. The authors claimed that there is a correlation between waste handling and the age (line 170) of the respondents and recommended specific training for a certain age bracket. Though waste handling is mostly on the hand of the waste management office of the hospital to include the janitors, crews, and lab technicians, but it was not presented in the methodology the current role of the respondents in the hospital. In line 97, the authors mentioned the demographic profile of the respondents but still, it lacks information about the respondents' current role at DGMAH. For instance, of course, if the respondent is a clerk then most probably the clerk does not have full knowledge as to how the waste in the hospital must be handled because it is not aligned with her/his work in the hospital. The manuscript though used the term waste handler which connotes a meaning that all 34 respondents are working as waste handlers at DGMAH.

All tables should be revised.

Line 73. Should be DGMAH

Line 90 to 91. “Participants were recruited from November 11, 2019, until August 10, 2020…. “ When did the actual study start? Is it after August 10, 2020 (which is after the “recruitment” of participants? Does this further mean that the period starting November 11, 2019, to August 10, 2020, is dedicated to recruiting the participants?

Line 91. What is the importance of asking the participants to explain their daily duties to the researcher as a basis for the selection process?

Lines 117 to 118. I do not understand this sentence.

Line 134 to 135. What particular work is referred to in these lines?

Line 159. In Table 2, what is meant by the column “all correct response, one correct response, and incorrect response”. Maybe the table can be restructured in a more informative way. There are some entries such as “Isolation waste must be labelled or marked isolation” that is confusing.

Line 162 to 163. Do the authors mean, that the respondents’ age, gender, education, or experience are not statistically related to their knowledge or ability of waste handling?

Line 165. “Knowledge of waste handlers regarding the types of waste generated in the isolation unit was not significantly influenced by these demographic and professional factors…”. Do the results of the survey, from 34 respondents, show that knowledge of handlers on waste handling is not statistically influenced by their age, gender, education, or experience however these finding seems not acceptable and is a contradiction of reality. I have a doubt that the number of respondents (34) is not enough to arrive at this general remark. Also, the scale of the response (0, 1, and 2) is very limited which may have affected the proper response of the respondents.

Line 172 to 173. The results show that the handlers’ age is not correlated with color identification. This is opposite to the result presented in line 170, where the waste handlers' knowledge of waste isolation is associated with age.

Line 183. Kindly revise Table 3 and present it in a simple yet informative one.

Line 191 to 192. “….only one correct response and incorrect responses…” Not clear.

Line 197. May be “importance” rather than “necessity”.

Line 203. Same comments for Table 4 as with Table 2.

Line 243. The discussion percey pf the manuscript is only a repetition of the information given in the Results. It is suggested to merge these 2 sections.

6. PLOS authors have the option to publish the peer review history of their article (what does this mean?). If published, this will include your full peer review and any attached files.

Reviewer #1: No

---

## [Author Response · Author response to Decision Letter 0]

28 Dec 2023

The paper presented information regarding the respondents' (represented by 34 respondents) awareness of waste handling and waste identification at Dr. George Mukhari Academic Hospital (DGMAH), Gauteng, South Africa. The authors claimed that there is a correlation between waste handling and the age (line 170) of the respondents and recommended specific training for a certain age bracket. Though waste handling is mostly on the hand of the waste management office of the hospital to include the janitors, crews, and lab technicians, but it was not presented in the methodology the current role of the respondents in the hospital. In line 97, the authors mentioned the demographic profile of the respondents but still, it lacks information about the respondents' current role at DGMAH. For instance, of course, if the respondent is a clerk then most probably the clerk does not have full knowledge as to how the waste in the hospital must be handled because it is not aligned with her/his work in the hospital. The manuscript though used the term waste handler which connotes a meaning that all 34 respondents are working as waste handlers at DGMAH

 All waste handlers are involved in the collection of isolation waste. This has been included in the methodology as part of the purposive sampling (Page 4). 

All tables should be revised.

Line 73. Should be DGMAH

Line 90 to 91. “Participants were recruited from November 11, 2019, until August 10, 2020…. “ When did the actual study start? Is it after August 10, 2020 (which is after the “recruitment” of participants? Does this further mean that the period starting November 11, 2019, to August 10, 2020, is dedicated to recruiting the participants?

Line 91. What is the importance of asking the participants to explain their daily duties to the researcher as a basis for the selection process?

Lines 117 to 118. I do not understand this sentence.

Line 134 to 135. What particular work is referred to in these lines?

Line 159. In Table 2, what is meant by the column “all correct response, one correct response, and incorrect response”. Maybe the table can be restructured in a more informative way. 

There are some entries such as “Isolation waste must be labelled or marked isolation” that is confusing.

```Line 162 to 163. Do the authors mean, that the respondents’ age, gender, education, or experience are not statistically related to their knowledge or ability of waste handling?

Line 165. “Knowledge of waste handlers regarding the types of waste generated in the isolation unit was not significantly influenced by these demographic and professional factors…”. Do the results of the survey, from 34 respondents, show that knowledge of handlers on waste handling is not statistically influenced by their age, gender, education, or experience however these finding seems not acceptable and is a contradiction of reality. I have a doubt that the number of respondents (34) is not enough to arrive at this general remark. 

Also, the scale of the response (0, 1, and 2) is very limited which may have affected the proper response of the respondents.

Line 172 to 173. The results show that the handlers’ age is not correlated with color identification. This is opposite to the result presented in line 170, where the waste handlers' knowledge of waste isolation is associated with age.

Line 183. Kindly revise Table 3 and present it in a simple yet informative one.

Line 191 to 192. “….only one correct response and incorrect responses…” Not clear.

Line 197. May be “importance” rather than “necessity”.

Line 203. Same comments for Table 4 as with Table 2.

Line 243. The discussion percey pf the manuscript is only a repetition of the information given in the Results. It is suggested to merge these 2 sections. 

This have all been corrected. 

Participants were recruited from November 11, 2019, until 4 December 2019.

Participants were asked to explain their daily duties to the researcher before taking part in the study the process was important to identify individuals who not only meet the criteria for the study.

This is the participants years of experience as a waste handler. 

In Table 2, the coding of the response was structured in a way that the respondents 

Isolation waste must be labelled or marked with the term isolation waste.

Yes. 

The entire waste handlers in the hospital were 34. The survey included all the waste handlers. As such, the 34 respondents are representative of the entire waste handlers involved in the collection of isolation waste. 

The result is not contradictory. While there was no association between age and knowledge of sharp objectives, and colour identification, there was, however, statistically significant difference with respect to their knowledge of isolation waste.

There is no other way to present Table 3 as it relates to association between the demographic and specific questions addressing their knowledge. 

However, there were 7 respondents (20.6%) who provided only one correct response despite being allowed to select multiple answers. This tendency may indicate a potential limitation in the depth of their knowledge. Furthermore, 2 participants (5.9%) provided incorrect responses to the question.

This has been corrected.

The responses were categorized as correct response- when they waste handlers answers all question questions, incorrect- when they failed to answer any of the questions correctly, and one correct response- when they answered one correct response. 

The comment is noted. The discussion sections discuss the results in relation to relevant literatures. The authors believe keeping both separate will be beneficial to the readers.

---

## [Decision Letter · Decision Letter 1]

29 Feb 2024

PONE-D-23-28839R1An investigation into waste handler’s knowledge of management of isolation waste: A case study of Dr George Mukhari Academic Hospital, Gauteng, South AfricaPLOS ONE

Dear Dr. Onwubu,

Thank you for submitting your manuscript to PLOS ONE. After careful consideration, we feel that it has merit but does not fully meet PLOS ONE’s publication criteria as it currently stands. Therefore, we invite you to submit a revised version of the manuscript that addresses the points raised during the review process.

**ACADEMIC EDITOR: **Please below comments, kindly address them.

We look forward to receiving your revised manuscript.

Kind regards,

Charles Odilichukwu R. Okpala

Academic Editor

PLOS ONE

**Additional Editor Comments:**

Please authors, kindly address the comments raised by reviewer. Thank you

Reviewers' comments:

Reviewer's Responses to Questions

**Comments to the Author**

1. If the authors have adequately addressed your comments raised in a previous round of review and you feel that this manuscript is now acceptable for publication, you may indicate that here to bypass the “Comments to the Author” section, enter your conflict of interest statement in the “Confidential to Editor” section, and submit your "Accept" recommendation.

Reviewer #2: All comments have been addressed

2. Is the manuscript technically sound, and do the data support the conclusions?

Reviewer #2: Partly

3. Has the statistical analysis been performed appropriately and rigorously? 

Reviewer #2: Yes

4. Have the authors made all data underlying the findings in their manuscript fully available?

Reviewer #2: Yes

5. Is the manuscript presented in an intelligible fashion and written in standard English?

Reviewer #2: No

6. Review Comments to the Author

Reviewer #2: Abstract

1. What were the specific findings regarding waste handlers' knowledge of waste types, color codes, safety precautions, and internal policies related to isolation waste management at Dr. George Mukhari Academic Hospital?

Introduction

1. What are the potential health and environmental risks associated with improper management of medical waste, specifically isolation waste?

2. Why is proper management of isolation waste crucial in healthcare facilities?

3. What are the critical steps involved in the management of isolation waste?

4. Are there specific guidelines and regulations in South Africa for healthcare waste management? Are these guidelines consistently followed by waste handlers in different healthcare settings?

5. What unique challenges related to waste management may Dr George Mukhari Academic Hospital face, and how do these challenges impact waste handlers' practices and compliance with isolation waste management protocols?

Methods

1. Can you provide more information about the selection process for participants in the study? How were waste handlers purposively selected?

2. How was permission obtained from the hospital management to contact the participants? Did the management also assist in the recruitment process?

3. Can you explain the process of data collection using the structured questionnaire? How were the questions formulated and what were the categories used for data variables?

4. How was the knowledge of waste handlers on handling isolation waste assessed? Can you provide more details about the rating scale used and the interpretation of the responses?

5. What were the ethical considerations taken into account for this study? Can you elaborate on the voluntary participation and the informed consent process?

Results

1. Can you provide more details on the methodology used to assess waste handlers' knowledge and understanding of isolation waste management?

2. Were there any limitations or challenges encountered during the study that may have affected the results?

3. What are the implications of waste handlers' good understanding of the types of waste generated in the isolation unit and the specific examples of sharp waste?

4. How does the correct identification of the colour code used to represent infectious waste contribute to proper waste segregation and disposal practices?

5. Can you explain the significance of waste categorization and disposal practices for environmental protection and sustainability goals in South Africa?

Conclusion

1. What specific training methods or strategies do you recommend for different age groups to improve their knowledge of isolation waste?

2. How do you suggest utilizing experienced waste handlers as mentors to enhance knowledge dissemination among newer staff members?

3. Are there any specific recommendations or actions that you propose for Dr. George Mukhari Academic Hospital to further improve waste management practices and ensure a safe working environment?

7. PLOS authors have the option to publish the peer review history of their article (what does this mean?). If published, this will include your full peer review and any attached files.

Reviewer #2: **Yes: **Dr. Debajyoti Kundu

---

## [Author Response · Author response to Decision Letter 1]

16 Mar 2024

1. What were the specific findings regarding waste handlers' knowledge of waste types, color codes, safety precautions, and internal policies related to isolation waste management at Dr. George Mukhari Academic Hospital?

Response:

We do not understand in clear terms the reviewer comments. However, the findings are included in the abstract

The study found that the majority of waste handlers demonstrated a good understanding of waste types generated in the isolation unit, including sharps waste, human tissue waste, infectious waste, and general waste. They also correctly identified examples of sharp waste, such as injections, blades, glass slides, and needles. Additionally, most respondents were aware of the colour code used for representing infectious waste as "yellow" and "red." The study revealed a statistically significant association between waste handlers' age and their knowledge of isolation waste, suggesting that age may influence their understanding of waste management practices. Furthermore, experience was found to be significantly associated with waste handlers' knowledge of the health-hazardous nature of isolation waste. While the majority of waste handlers recognized the importance of wearing protective clothing and correctly marking isolation waste, some respondents were not aware of the internal policy for waste handling such as guidelines and protocols specific to the segregation, packaging, labeling, and disposal of waste generated within the isolation units (Page 1). 

What are the potential health and environmental risks associated with improper management of medical waste, specifically isolation waste?

Response:

This has been provided: 

Improper disposal of isolation waste presents significant risks to both healthcare workers and the environment. For instance, inadequate disposal of sharps waste, including needles and syringes, can lead to needlestick injuries among healthcare workers. Such injuries pose a serious risk of exposure to bloodborne pathogens such as HIV, hepatitis B, and hepatitis C (5). Similarly, improper disposal of isolation waste, such as dumping or inadequate landfilling, can have detrimental effects on the environment. Contaminating soil and groundwater with hazardous chemicals and pathogens from isolation waste poses risks to ecosystems and human health (6). (Page 2). 

What are the critical steps involved in the management of isolation waste?

Response:

These are explained in page 2 line 57. For reference: The management of isolation waste involves several critical steps, including waste segregation, proper containment, safe transportation, and appropriate disposal (7).

Are there specific guidelines and regulations in South Africa for healthcare waste management? Are these guidelines consistently followed by waste handlers in different healthcare settings?

Response:

Yes, there are specific guidelines and regulations of medical waste in South Africa (The management of Healthcare Risk Waste (HCRW) is regulated by various legislation, including the National Health Act no. 61 of 2003, the National Environmental Management Act no. 107 of 1998, the National Environmental Management Waste Act no. 59 of 2008, and National Core Standards (10). These regulations establish comprehensive protocols governing the handling, storage, collection, transportation, treatment, and disposal of medical waste (Page 3).

However, these guidelines are not consistently followed by waste handles (Page 3).

What unique challenges related to waste management may Dr George Mukhari Academic Hospital face, and how do these challenges impact waste handlers' practices and compliance with isolation waste management protocols?

Response:

Given that compliance remains a challenge in many hospitals across South Africa, we assume there may be variations in knowledge, awareness, and adherence to waste handling guidelines among waste handlers in different healthcare settings including Dr George Mukhari is no exception. Hence, the need to investigate the level of knowledge and awareness of isolation waste among handlers in Dr George Muhkari Academic Hospital. This was informed as the hospital serves as a significant tertiary medical institution in the region. As a facility providing healthcare services to a diverse patient population, including those with infectious diseases. 

Can you provide more information about the selection process for participants in the study? How were waste handlers purposively selected?

Response:

hospital management played a crucial role in participant recruitment by disseminating information about the study to the heads of departments and waste handlers in their respective departments. Participants were purposively selected from waste handlers stationed in hospital wards, specifically those responsible for handling isolation waste, including its removal, packaging, and sealing (Page 4).

How was permission obtained from the hospital management to contact the participants? Did the management also assist in the recruitment process?

Response:

Yes, the hospital management played a crucial role in participant recruitment by disseminating information about the study to the heads of departments and waste handlers in their respective departments (Page 4). 

Can you explain the process of data collection using the structured questionnaire? How were the questions formulated and what were the categories used for data variables?

Response:

These are explained as: Data collection was conducted using a structured questionnaire comprising close-ended questions. The use of closed questions facilitated easy processing of answers, enhanced comparability, and facilitated the presentation of relationships between variables, allowing for comparisons between respondents. The questionnaire consisted of four parts, and data variables were categorized accordingly:

● Part one covered the demographics (age, gender, language, and highest qualifications).

● Part two covered knowledge of isolation waste.

● Part three focused on the understanding of risks associated with handling isolation waste. 

● Part four looked at training and policies on isolation waste.

How was the knowledge of waste handlers on handling isolation waste assessed? Can you provide more details about the rating scale used and the interpretation of the responses?

Response: 

To assess the knowledge of waste handlers on handling isolation waste, a rating scale was used whereby responses were rated from 0-2, with 0 indicating very poor, 1 being good and 2 being very good. To respond to questions on the usage of hygiene, PPE, safety precautions and training, respondents had to select the correct answer and three or five options were listed, some questions required respondents to give a yes/no/ do not know response (Page 5).

What were the ethical considerations taken into account for this study? Can you elaborate on the voluntary participation and the informed consent process?

Response:

Ethical clearance for this study was obtained from the Institutional Research Ethics Committee at Durban University of Technology, with the assigned IREC number 139/19. Subsequently, permission to conduct the study and contact the participants was sought and granted from the Gauteng Department of Health. Participation in the study was entirely voluntary, and participants had the right to withdraw at any point without facing any consequences. Those who agreed to take part were provided with consent forms and letters of information detailing the study's purpose, procedures, and their rights as participants. To ensure confidentiality, the researcher refrained from using participants' names in the report. Additionally, no information or data that could potentially identify respondents was included. The collected information was solely used for the research report and was not shared for any other purpose. Participants were also assured that their involvement in the study would not result in harm or injury. Moreover, the potential benefits of the study were explained, emphasizing its contribution to improving isolation waste management practices in healthcare facilities (Page 6).

Can you provide more details on the methodology used to assess waste handlers' knowledge and understanding of isolation waste management?

Response:

The knowledge and understanding of isolation waste were assessed using structured questionnaire. All questions that allowed respondents to select more than one correct answer were assigned a score of 2 if the respondent provided multiple correct answers. For questions with single correct responses answered correctly, a score of 1 was awarded. Incorrect or negative responses were scored as 0.

Were there any limitations or challenges encountered during the study that may have affected the results?

Response: 

None was encountered. 

What are the implications of waste handlers' good understanding of the types of waste generated in the isolation unit and the specific examples of sharp waste?

Response:

These are provided in the discussion section. For reference: The findings revealed that a majority of waste handlers demonstrated a good understanding of the types of waste generated in the isolation unit, as well as the specific examples of sharp waste, such as injections, blades, glass slides, and needles. This is practically important as the understanding of waste handlers in the handling of isolation waste could contribute to a safer and healthier environment within the healthcare facility. Besides, the understanding the types of waste, especially sharp waste, ensures that waste handlers can take appropriate safety precautions to prevent injuries and potential exposure to harmful pathogens. Proper handling and disposal of sharp waste reduce the risk of needlestick injuries and transmission of bloodborne infections such as HIV, hepatitis B, and hepatitis C. This aligns with Sutrisno and Meilasari (16) that effective management of the waste generated is of vital importance to curb the spread of disease (Page 14). 

How does the correct identification of the colour code used to represent infectious waste contribute to proper waste segregation and disposal practices?

Response:

The proper categorization and segregation of different types of waste, such as hazardous and non-hazardous waste, and the implementation of appropriate disposal methods, are highly significant for South Africa. These practices can minimize the release of harmful pollutants into the environment, which is crucial for protecting ecosystems and safeguarding human health (17). In addition, waste categorization and disposal practices are essential components of South Africa's efforts to achieve environmental protection and sustainability goals, as outlined in the legislative framework (19).

What specific training methods or strategies do you recommend for different age groups to improve their knowledge of isolation waste?

Response:

Training initiatives could include on-the-job training sessions designed for healthcare workers and waste handlers to familiarize them with isolation waste management protocols, safety procedures, and regulatory requirements. Additionally, the provision of continuing education programs or professional development courses focusing on isolation waste management topics, such as infection control, hazardous waste handling, and personal protective equipment (PPE) usage, is essential. These initiatives aim to reinforce waste management practices and promote adherence to safety protocols.

How do you suggest utilizing experienced waste handlers as mentors to enhance knowledge dissemination among newer staff members?

Response:

To achieve this, formal mentorship programs can be established, pairing experienced waste handlers with newer staff members based on job roles, areas of expertise, or specific needs identified during training.

Are there any specific recommendations or actions that you propose for Dr. George Mukhari Academic Hospital to further improve waste management practices and ensure a safe working environment?

Response:

Overall, the study recommends that Dr. George Mukhari Academic Hospital conduct regular training sessions and workshops for healthcare workers and waste handlers on proper waste segregation, handling, and disposal procedures. This includes providing education on infection control measures, instructing on the proper use of personal protective equipment (PPE), and ensuring compliance with regulatory requirements. Additionally, hospital management should regularly monitor and supervise waste management practices to ensure compliance with protocols and regulations. By following these recommendations, Dr. George Mukhari Academic Hospital can further improve waste management practices, ensure a safe working environment, and contribute to infection control and environmental protection.

---

## [Decision Letter · Decision Letter 2]

6 May 2024

PONE-D-23-28839R2An investigation into waste handler’s knowledge of management of isolation waste: A case study of Dr George Mukhari Academic Hospital, Gauteng, South AfricaPLOS ONE

Dear Dr. Onwubu,

Thank you for submitting your manuscript to PLOS ONE. After careful consideration, we feel that it has merit but does not fully meet PLOS ONE’s publication criteria as it currently stands. Therefore, we invite you to submit a revised version of the manuscript that addresses the points raised during the review process.

We look forward to receiving your revised manuscript.

Kind regards,

Charles Odilichukwu R. Okpala

Academic Editor

PLOS ONE

Journal Requirements:

Additional Editor Comments:

Please, kindly attend to the minor comments

Reviewers' comments:

Reviewer's Responses to Questions

**Comments to the Author**

1. If the authors have adequately addressed your comments raised in a previous round of review and you feel that this manuscript is now acceptable for publication, you may indicate that here to bypass the “Comments to the Author” section, enter your conflict of interest statement in the “Confidential to Editor” section, and submit your "Accept" recommendation.

Reviewer #2: All comments have been addressed

Reviewer #3: (No Response)

2. Is the manuscript technically sound, and do the data support the conclusions?

Reviewer #2: Yes

Reviewer #3: Yes

3. Has the statistical analysis been performed appropriately and rigorously? 

Reviewer #2: N/A

Reviewer #3: Yes

4. Have the authors made all data underlying the findings in their manuscript fully available?

Reviewer #2: Yes

Reviewer #3: Yes

5. Is the manuscript presented in an intelligible fashion and written in standard English?

Reviewer #2: Yes

Reviewer #3: Yes

6. Review Comments to the Author

Reviewer #2: (No Response)

Reviewer #3: (No Response)

7. PLOS authors have the option to publish the peer review history of their article (what does this mean?). If published, this will include your full peer review and any attached files.

Reviewer #2: No

Reviewer #3: No

---

## [Author Response · Author response to Decision Letter 2]

11 May 2024

1 Data collection was conducted using a structured questionnaire comprising close-ended questions. The questionnaire consisted of four parts, and data variables were categorized accordingly: This might be present in a flow chart diagram. Lines 122 – 125; Page 5

Response

This has been provided (Page 5). 

The reviewer strongly encourages the Authors to include the waste segregation system and collection scheme applied in the Country of study (South Africa) in the introduction section. This will add quality to their finding and justify the study.

Response

This has been included (Page 3 & 4). 

The reviewer acknowledges the primary focus of the study was the investigation into waste handler’s knowledge of the management of isolation waste. However, do Authors consider the treatment of health care waste such as biological treatment, Incinerating treatment, Steam-based treatment technologies, etc. might also be a solution to the challenges in South Africa? 

Response

The comment is appreciated. However, we did not investigate the waste treatment process as it is beyond the scope to the present study. Our study primary focus on waste handlers’ knowledge. We, however, may consider the suggestion as a future study. 

Lines: 166 – 167 “In terms of age groups, 50% were between 40-49, 23.5% each were below 20 and between 21-39, and the same for 50 and below” Please re-write this part (40-49, 23.5%) a bit difficult to understand. A similar situation is found in the result section. 

Response

This has been corrected and revised as: 

Socio-demographic characteristics 

The socio-demographic characteristic of the waste handlers is given in Table 1. The gender distribution showed 85.3% females and 14.7% males. In terms of age groups, half of the respondents (50%) were between 40-49. Education-wise, most of the respondents (47.1%) completed high school while those who had some high schooling makes up 38.2% of sample. In terms of their experience as a waste handler, most of the respondents (47.1%) had 6-11 years (Page 7).

Please be consistent with punctuation: line 176 Author wrote ((n=25, 73.5%), I think instead of (n=25; 73.5%). A similar situation is found in the result section.

Response

This has been corrected. 

The reviewer strongly recommends Authors improve Table 3. (Give a space in between rows associated with each list of Knowledge on isolation unit waste). It is very important to readers for clarity. For instance, Table 2. is a good example of how tables should be. 

Response

This has been addressed as per the recommendation (Page 11).

Please revise Table 2: Knowledge of waste handlers on different types of waste generated in isolation unit, if it is a mistake. “Knowledge of isolation waste” is presented twice in the first column This has bee revised (Page 9). 

Recommend Authors for more depth of discussion. For instance, in lines 299 - 300 (According to (17), adequate waste categorization and disposal practices have implications for environmental protection). Please compare your findings with some of those references in your manuscript with what exactly they do. This will increase the quality of your study. 

Response

The comment is noted. However, the supporting reference cited was to clarify how correct identification of colour code used to represent infectious waste contribute to proper waste segregation and disposal practices. The following has also been added to strengthen the discussion: The findings is consistent with Senekane and Masimula (11) that 90% medical practitioners generated waste and disposed of into bins lined with red plastics, 2.5% disposed of into yellow plastic bags.

---

## [Decision Letter · Decision Letter 3]

28 May 2024

An investigation into waste handler’s knowledge of management of isolation waste: A case study of Dr George Mukhari Academic Hospital, Gauteng, South Africa

PONE-D-23-28839R3

Dear Dr. Onwubu,

We’re pleased to inform you that your manuscript has been judged scientifically suitable for publication and will be formally accepted for publication once it meets all outstanding technical requirements.

Kind regards,

Charles Odilichukwu R. Okpala

Academic Editor

PLOS ONE

Additional Editor Comments (optional):

This revised manuscript is worthy of publication.

Reviewers' comments:

Reviewer's Responses to Questions

**Comments to the Author**

1. If the authors have adequately addressed your comments raised in a previous round of review and you feel that this manuscript is now acceptable for publication, you may indicate that here to bypass the “Comments to the Author” section, enter your conflict of interest statement in the “Confidential to Editor” section, and submit your "Accept" recommendation.

Reviewer #3: All comments have been addressed

2. Is the manuscript technically sound, and do the data support the conclusions?

Reviewer #3: Yes

3. Has the statistical analysis been performed appropriately and rigorously? 

Reviewer #3: Yes

4. Have the authors made all data underlying the findings in their manuscript fully available?

Reviewer #3: Yes

5. Is the manuscript presented in an intelligible fashion and written in standard English?

Reviewer #3: Yes

6. Review Comments to the Author

Reviewer #3: (No Response)

7. PLOS authors have the option to publish the peer review history of their article (what does this mean?). If published, this will include your full peer review and any attached files.

Reviewer #3: No

---

## [Editor Report · Acceptance letter]

30 May 2024

PONE-D-23-28839R3 

PLOS ONE

Dear Dr. Onwubu, 

I'm pleased to inform you that your manuscript has been deemed suitable for publication in PLOS ONE. Congratulations! Your manuscript is now being handed over to our production team.

Kind regards, 

on behalf of

Dr. Charles Odilichukwu R. Okpala 

Academic Editor

PLOS ONE